# T1: Scaling Diffusion Probabilistic Fields to High-Resolution on Unified Visual Modalities

## Abstract

Diffusion Probabilistic Field (DPF) (Zhuang et al., 2023) models the distribution of continuous functions defined over metric spaces. While DPF shows great potential for unifying data generation of various modalities including images, videos, and 3D geometry, it does not scale to a higher data resolution. This can be attributed to the "scaling property", where it is difficult for the model to capture local structures through uniform sampling. To this end, we propose a new model comprising of a view-wise sampling algorithm to focus on local structure learning, and incorporating additional guidance, *e.g.*, text description, to complement the global geometry. The model can be scaled to generate high-resolution data while unifying multiple modalities. Experimental results on data generation in various modalities demonstrate the effectiveness of our model, as well as its potential as a foundation framework for scalable modality-unified visual content generation.

## 1 Introduction

Generative tasks (Rombach et al., 2022; Ramesh et al., 2022) are overwhelmed by diffusion probabilistic models that hold state-of-the-art results on most modalities like audio, images, videos, and 3D geometry. Take image generation as an example, a typical diffusion model (Ho et al., 2020) consists of a forward process for sequentially corrupting an image into standard noise, a backward process for sequentially denoising a noisy image into a clear image, and a score network that learns to denoise the noisy image.

The forward and backward processes are agnostic to different data modalities; however, the architectures of the existing score networks are not. The existing score networks are highly customized towards a single type of modality, which is challenging to adapt to a different modality. For example, a recently proposed multi-frame video generation network (Ho et al., 2022b;a) adapting single-frame image generation networks involves significant designs and efforts in modifying the score networks. Therefore, it is important to develop a unified model that works across various modalities without modality-specific customization, in order to extend the success of diffusion models across a wide range of scientific and engineering disciplines, like medical imaging (*e.g.*, MRI, CT scans) and remote sensing (*e.g.*, LiDAR).

Field model (Sitzmann et al., 2020; Tancik et al., 2020; Dupont et al., 2022b; Zhuang et al., 2023) is a promising unified score network architecture for different modalities. It learns the distribution over the functional view of data. Specifically, the field $f$ maps the observation from the *metric* space $\mathcal{M}$ (*e.g.*, coordinate or camera pose) into the *signal* space $\mathcal{Y}$ (*e.g.*, RGB pixel) as $f : \mathcal{M} \mapsto \mathcal{Y}$. For instance, an image is represented as $f : \mathbb{R}^2 \mapsto \mathbb{R}^3$ that maps the spatial coordinates (*i.e.*, height and width) into RGB values at the corresponding location (See Fig. 1 (a)), while a video is represented as $f : \mathbb{R}^3 \mapsto \mathbb{R}^3$ that maps the spatial and temporal coordinates (*i.e.*, frame, height, and width) into RGB values (See Fig. 1 (b)). Recently, diffusion models are leveraged to characterize the field distributions over the functional view of data (Zhuang et al., 2023) for field generation. Given a set of coordinate-signal pairs $\{(\boldsymbol{m}_i, \boldsymbol{y}_i)\}$, the field $f$ is regarded as the score network for the backward process, which turns a noisy signal into a clear signal $\boldsymbol{y}_i$ in a sequential process with $\boldsymbol{m}_i$ being fixed all the time, as shown in Fig. 1 (d). The visual content is then composed of the clear signal generated on a grid in the metric space.

Nevertheless, diffusion-based field models for generation still lag behind the modality-specific approaches (Dhariwal & Nichol, 2021; Ho et al., 2022b; He et al., 2022) for learning from dynamic

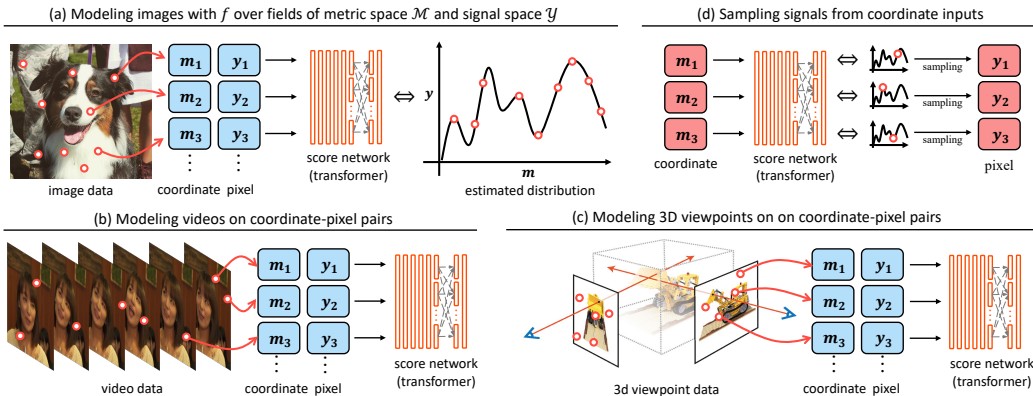

Figure 1: Illustration of the field models' capability of modeling visual content. The underlying data distribution is simplified into the 1-D space for demonstration. The score network learns the distribution through the attention among coordinate-signal pairs, which is modality-agnostic.

data in high resolution (Bain et al., 2021; Yu et al., 2023a). For example, a 240p video lasting 5 seconds is comprised of up to 10 million coordinate-signal pairs. Due to the memory bottleneck in existing GPU-accelerated computing systems, recent field models (Zhuang et al., 2023) are limited to observe merely a small portion of these pairs (*e.g.*, $1\%$) that are uniformly sampled during training. This limitation significantly hampers the field models in approximating distributions from such sparse observations (Quinonero-Candela & Rasmussen, 2005). Consequently, diffusion-based field models often struggle to capture the fine-grained local structure of the data, leading to, *e.g.*, unsatisfactory blurry results.

While it is possible to change the pair sampling algorithm to sample densely from local areas instead of uniformly, the global geometry is weakened. To alleviate this issue, it is desirable to introduce some complementary guidance on the global geometry in addition to local sampling.

Multiple attempts (Gordon et al., 2020; Dutordoir et al., 2022; Zhuang et al., 2023) have been presented to introduce additional global priors during modeling. Recent diffusion models (Rombach et al., 2022; Ramesh et al., 2022) demonstrate that text descriptions can act as strong inductive biases for learning data distributions, by introducing global geometry priors of the data, thereby helping one to scale the models on complex datasets. However, fully exploiting correlation between the text and the partially represented field remains uncharted in the literature.

In this paper, we aim to address the aforementioned issues, and scale the field models for generating high-resolution, dynamic data. We propose a new diffusion-based field model, called **T1**. In contrast to previous methods, T1 preserves both the local structure and the global geometry of the fields during learning by employing a new view-wise sampling algorithm in the coordinate space, and incorporates additional inductive biases from the text descriptions. By combining these advancements with our simplified network architecture, we demonstrate that T1's modeling capability surpasses previous methods, achieving improved generated results under the same memory constraints. We empirically validate its superiority against previous domain-agnostic methods across three different tasks, including image generation, text-to-video generation, and 3D viewpoint generation. Various experiments show that T1 achieves compelling performance even when compared to the state-of-the-art domain-specific methods, underlining its potential as a scalable and unified visual content generation model across various modalities. Notably, T1 is capable of generating high-resolution video under affordable computing resources, while the existing field models can not.

Our contributions are summarized as follows:

- We reveal the scaling property of diffusion-based field models, which prevents them from scaling to high-resolution, dynamic data despite their capability of unifying various visual modalities.
- We propose T1, a new diffusion-based field model with a sampling algorithm that maintains the view-wise consistency, and enables the incorporation of additional inductive biases.

## 2  BACKGROUND

Conceptually, the diffusion-based field models sample from field distributions by reversing a gradual noising process. As shown in Fig. 1, in contrast to the data formulation of the conventional diffusion models (Ho et al., 2020) applied to the complete data like a whole image, diffusion-based field models apply the noising process to the sparse observation of the field, which is a kind of parametrized functional representation of data consisting of coordinate-signal pairs, *i.e.*, $f : \mathcal{M} \mapsto \mathcal{Y}$. Specifically, the sampling process begins with a coordinate-signal pair $(\mathbf{m}_i, \mathbf{y}_{(i,T)})$, where the coordinate comes from a field and the signal is a standard noise, and less-noisy signals $\mathbf{y}_{(i,T-1)}, \mathbf{y}_{(i,T-2)}, \ldots,$ are progressively generated until reaching the final clear signal $\mathbf{y}_{(i,0)}$, with $\mathbf{m}_i$ being constant.

Diffusion Probabilistic Field (DPF) Zhuang et al. (2023) is one of the recent representative diffusion-based field models. It parameterizes the denoising process with a transformer-based network $\epsilon_\theta(\cdot)$, which takes noisy coordinate-signal pairs as input and predicts the noise component $\epsilon$ of $\mathbf{y}_{(i,t)}$. The less-noisy signal $\mathbf{y}_{(i,t-1)}$ is then sampled from the noise component $\epsilon$ using a denoising process Ho et al. (2020). For training, they use a simplified loss proposed by Ho et al. (Ho et al., 2020) instead of the variational lower bound for modeling the distributions in VAE (Kingma & Welling, 2013). Specifically, it is a simple mean-squared error between the true noise and the predicted noise, *i.e.*, $\|\epsilon_\theta(\mathbf{m}_i, \mathbf{y}_{(i,t)}, t) - \epsilon\|$. This approach is found better in practice and is equivalent to the denoising score matching model (Song & Ermon, 2020), which belongs to another family of denoising models and is referred to as the denoising diffusion model.

In practice, when handling low-resolution data consisting of $N$ coordinate-signal pairs with DPF, the scoring network $\epsilon_\theta(\cdot)$ takes all pairs $\{(\mathbf{m}_i, \mathbf{y}_{(i,T)})\}$ as input at once. For high-resolution data with a large number of coordinate-signal pairs that greatly exceed the modern GPU capacity, Zhuang et al. (2023) uniformly sample a subset of pairs from the data as input. They subsequently condition the diffusion model on the other non-overlapping subset, referred to as *context pairs*. Specifically, the sampled pairs interact with the query pairs through cross-attention blocks. Zhuang et al. (2023) show that the ratio between the context pairs and the sampling pairs is strongly related to the quality of the generated fields, and the quality decreases as the context pair ratio decreases. Due to the practical memory bottleneck, DPF can only support a maximum $64 \times 64$ resolution, but our method can handle a larger resolution $256 \times 256 \times 128$ with the same hardware. In particular, our proposed field model operates on the pairs constructed in the latent space of the autoencoder, instead of the pairs constructed on the raw pixel space. As a result, while our method and DPF take the same number of pairs as input, our method naturally allows dealing with views in higher resolution.

## 3  METHOD

In order to scale diffusion-based field models for high-resolution, dynamic data generation, we build upon the recent DPF model (Zhuang et al., 2023) and address its limitations in preserving the local structure of fields, as it can hardly be captured when the uniformly sampled coordinate-signal pairs are too sparse. Specially, our method not only can preserve the local structure, but also enables introducing additional inductive biases (*i.e., text descriptions)* for capturing the global geometry.

### 3.1  VIEW-WISE SAMPLING ALGORITHM

In order to preserve the local structure of fields, we propose a new view-wise sampling algorithm that samples local coordinate-signal pairs for better representing the local structure of fields. For instance, the algorithm samples the coordinate-signal pairs belonging to a single or several ($n \geqslant 1$; $n$ denotes the number of views) views for video data, where a view corresponds to a single frame. It samples pairs belonging to a single or several rendered images for 3D viewpoints, where a view corresponds to an image rendered at a specific camera pose. A view of an image is the image itself.

This approach restricts the number of interactions among pairs to be modeled and reduces the learning difficulty on high-resolution, dynamic data. Nevertheless, even a single high-resolution view , *e.g.*, in merely $128 \times 128$ resolution) can still consist of 10K pairs, which in practice will very easily reach the memory bottleneck if we leverage a large portion of them at one time, and hence hinder scaling the model for generating high-resolution dynamic data.

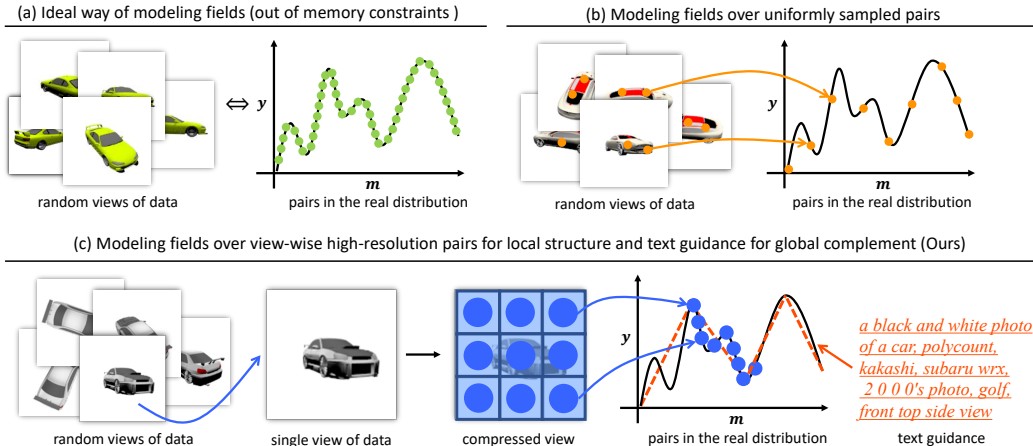

Figure 2: Sampling strategies on high-dimensional data. (a) Ideally, all pairs within a field (green points) should be used for training, but is impractical given the memory limitations. (b) Previous methods uniformly sample a sparse set of pairs (orange points) to represent the field. (c) Compared to uniform sampling, our local sampling extracts high-resolution pairs (blue points), better covering the local structure. The text guidance (red line) as an approximation complements the global geometry.

To address this issue, our method begins by increasing the signal resolution of coordinate-signal pairs and reducing memory usage in the score network. Specifically, we replace the signal space with a compressed latent space, and employ a more efficient network architecture that only contains decoders. This improvement in efficiency allows the modeling of interactions among pairs representing higher-resolution data while keeping the memory usage constrained. Based on this, one can then model the interactions of pairs within a single or several views of high-resolution data. The overall diagram of the proposed sampling algorithm can be found in Fig. 2.

**Signal Resolution.**    We construct the coordinate-signal pairs in a compressed latent space, *i.e.*, each signal is represented by a transformer token, where the signal resolution for each token is increased from $1 \times 1 \times 3$ to $16 \times 16 \times 3$ compared to the baseline, while maintaining the memory cost of each pair. In particular, for each view of the data in a $H \times W \times 3$ resolution, we first extract its latent representation using a pre-trained autoencoder (Rombach et al., 2022), with the latent map size being $H/8 \times W/8 \times 4$. This approach improves the field representation efficiency by perceptually compressing the resolution. We then employ a convolutional layer with $2 \times 2$ kernel size in the score network for further compressing the latent, resulting in a compressed feature map in $H/16 \times H/16 \times 4$ resolution. This step further improves the computation efficiency of the scoring network by four times, which is particularly useful for transformers that have quadratic complexity.

In this way, each coordinate-signal pair contains a coordinate, and its corresponding $1 \times 1$ feature point (corresponds to a $16 \times 16$ signal) from the compressed feature map (with positional embedding added). For each token, we use their corresponding feature map location for the position embedding. By combining these, in comparison to DPF which supports a maximum $64 \times 64$ view resolution, our method can handle views with a maximum resolution of $1024 \times 1024$ while maintaining very close memory consumption during learning without compromising the quality of the generated signal.

**Score Network.**    We further find that once a token encapsulates enough information to partially represent the fidelity of the field, the context pairs (Zhuang et al., 2023) are no longer necessary for model efficiency. Therefore, using high-resolution tokens enables us to get rid of the encoder-decoder architecture (Jaegle et al., 2021) and thus to utilize a more parameters-efficient decoder-only architecture. We adopt DiT (Peebles & Xie, 2022a) as the score network, which is the first decoder-only pure-transformer model that takes noisy tokens and positional embedding as input and generates the less-noisy token.

**View-wise Sampling Algorithm.**    Based on the high-resolution signal and decoder-only network architecture, our method represents field distributions by using view-consistent coordinate-signal

pairs, *i.e.*, collections of pairs that belong to a single or several ($n \geqslant 1$) views of the data, such as one or several frames in a video, and one or several viewpoints of a 3D geometry. In particular, take the spatial and temporal coordinates of a video in $H \times W$ resolution lasting for $T$ frames as an example, for all coordinates $\{\mathbf{m}_1, \mathbf{m}_2, \ldots, \mathbf{m}_i, \ldots, \mathbf{m}_{H \times W \times T}\}$, we randomly sample a consecutive sequence of length $H \times W$ that correspond to a single frame, *i.e.*, $\{\mathbf{m}_1, \mathbf{m}_2, \ldots, \mathbf{m}_i, \ldots, \mathbf{m}_{H \times W}\}$. For data consisting of a large amount of views (*e.g.* $T >> 16$), we randomly sample $n$ views (sequences of length $H \times W$), resulting in an $H \times W \times n$ sequence set. Accordingly, different from the transformers in previous works (Zhuang et al., 2023) that model interaction among all pairs across all views, ours only models the interaction among pairs that belongs to the same view, which reduces the complexity of field model by limiting the number of interactions to be learned.

## 3.2 TEXT CONDITIONING

To complement our effort in preserving local structures that may weaken global geometry learning, since the network only models the interaction of coordinate-signal pairs in the same view, we propose to supplement the learning with a coarse global approximation of the field, avoiding issues in cross-view consistency like worse spatial-temporal consistency between frames in video generation.

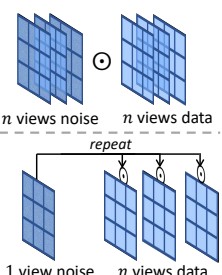

*n* views noise    *n* views data

*repeat*

1 view noise    *n* views data

Figure 3: Overview of the previous noisy data construction (top) and ours (bottom).

In particular, we propose to condition diffusion models on text descriptions related to the fields. Compared with the other possible modalities, text can better represent data in compact but highly expressive features (Devlin et al., 2018; Brown et al., 2020; Raffel et al., 2020), and serve as a low-rank approximation of data (Radford et al., 2021). By conditioning diffusion models on text descriptions, we show our method can capture the global geometry of data from texts. It works like inductive biases of each pairs and allow us to model cross-view interactions of pairs without explicit cross-attention used in previous methods (Zhuang et al., 2023).

**Cross-view Noise Consistency.** We propose to model the interactions among pairs across different views, which indeed represent the dependency between views as the global geometry. In particular, we perform the forward diffusion process that constructs cross-view noisy pairs by using the same noise component across views, as illustrated in Fig. 3. The reparameterization trick (Kingma & Welling, 2013) (for the forward process) is then applied to a set of sampled pairs $\mathbf{Q}$ of a field, where the pairs make up multiple views, as shown below:

$$
\begin{aligned}
\mathbf{Q} &= \left\{ \underbrace{\{(\mathbf{m}_i, \mathbf{y}_{(i,t)}) | i = 1, 2, \ldots, H{\cdot}W\}}_{\text{pairs from the } n\text{-th view}} \;\Big|\; n = 1, 2, \ldots, N \right\} \\
&= \left\{ \{(\mathbf{m}_{(i,n)}, \mathbf{y}_{(i,n,t)} = \sqrt{\bar{\alpha}}\mathbf{y}_{(i,n,0)} + \sqrt{1 - \bar{\alpha}_t}\epsilon_i) | i = 1, 2, \ldots, H{\cdot}W\} \;\Big|\; n = 1, 2, \ldots, N \right\}.
\end{aligned}
\tag{1}
$$

In contrast to the previous works that use different noise components for all views of a field, ours results in a modified learning objective, *i.e.*, to coherently predict the same noise component from different distorted noisy views. In this way, the whole field is regarded as a whole where each view is correlated with the others. This enforces the model to learn to generate coherent views of a field. During sampling, we use the deterministic DDIM sampler and only ensure the diffusion process started by the same noise component across views.

**Cross-view Condition Consistency.** In order to model the dependency variation between views belonging to the same field, *i.e.*, the global geometry of the field, we condition the diffusion model on the text embeddings of the field description or equivalent embeddings (*i.e.*, the language embedding of a single view in the CLIP latent space (Radford et al., 2021)). Our approach leverages the adaptive layer normalization layers in GANs (Brock et al., 2018; Karras et al., 2019), and adapts them by modeling the statistics from the text embeddings of shape $Z \times D$. For pairs that make up a single view, we condition on their represented tokens $Z \times D$, ($Z$ tokens of size $D$), by modulating them with the scale and shift parameters regressed from the text embeddings. For pairs $(T \times Z) \times D$ that make up multiple views, we condition on the view-level pairs by modulating feature in $Z \times D$ for each of the $T$ views with the same scale and shift parameters. Specifically,

| Model | CIFAR10 64×64 | | CelebV-Text 256×256×128 | | | ShapeNet-Cars 128×128×251 | | | |
|---|---|---|---|---|---|---|---|---|---|
| | FID (↓) | IS (↑) | FVD (↓) | FID (↓) | CLIPSIM (↑) | FID (↓) | LPIPS (↓) | PSNR (↑) | SSIM (↑) |
| Functa Dupont et al. (2022a) | 31.56 | 8.12 | ✗ | ✗ | ✗ | 80.30 | 0.183 | N/A | N/A |
| GEM Du et al. (2021) | 23.83 | 8.36 | ✗ | ✗ | ✗ | ✗ | ✗ | ✗ | ✗ |
| DPF Zhuang et al. (2023) | 15.10 | 8.43 | ✗ | ✗ | ✗ | 43.83 | 0.158 | 18.6 | 0.81 |
| TFGAN Balaji et al. (2019) | ✗ | ✗ | 571.34 | 784.93 | 0.154 | ✗ | ✗ | ✗ | ✗ |
| MMVID Han et al. (2022b) | ✗ | ✗ | 109.25 | 82.55 | 0.174 | ✗ | ✗ | ✗ | ✗ |
| MMVID-interp Han et al. (2022b) | ✗ | ✗ | 80.81 | 70.88 | 0.176 | ✗ | ✗ | ✗ | ✗ |
| VDM Ho et al. (2022b) | ✗ | ✗ | 81.44 | 90.28 | 0.162 | ✗ | ✗ | ✗ | ✗ |
| CogVideo Hong et al. (2023) | ✗ | ✗ | 99.28 | 54.05 | 0.186 | ✗ | ✗ | ✗ | ✗ |
| EG3D-PTI Chan et al. (2022) | ✗ | ✗ | ✗ | ✗ | ✗ | 20.82 | 0.146 | 19.0 | 0.85 |
| ViewFormer Kulhánek et al. (2022) | ✗ | ✗ | ✗ | ✗ | ✗ | 27.23 | 0.150 | 19.0 | 0.83 |
| pixelNeRF Yu et al. (2021) | ✗ | ✗ | ✗ | ✗ | ✗ | 65.83 | 0.146 | 23.2 | 0.90 |
| **T1 (Ours)** | **7.29** | **9.31** | **42.03** | **24.33** | **0.220** | **24.36** | **0.118** | **23.9** | **0.90** |

Table 1: Sample quality comparison with state-of-the-art models for each task. "✗" denotes the method cannot be adopted to the modality due to the method design or impractical computation cost.

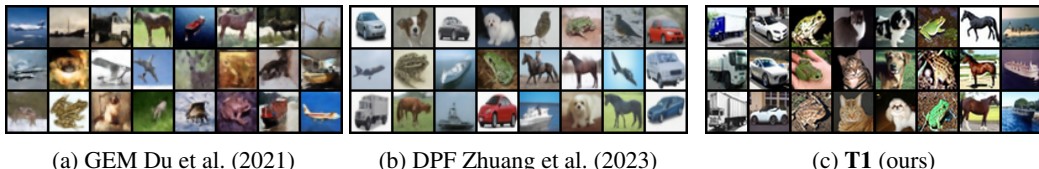

(a) GEM Du et al. (2021)  (b) DPF Zhuang et al. (2023)  (c) **T1** (ours)

Figure 4: Qualitative comparisons of domain-agnostic methods and ours on CIFAR-10. Our results show better visual quality with more details than the others, while being domain-agnostic as well.

each transformer blocks of our score network learns to predict statistic features $\beta_c$ and $\gamma_c$ from the text embeddings per channel. These statistic features then modulate the transformer features $F_c$ as:
$$\text{adLNorm}(F_c|\beta_c, \gamma_c) = \text{Norm}(F_c) \cdot \beta_c + \beta_c.$$

## 4 EXPERIMENTAL RESULTS

We demonstrate the effectiveness of our method on multiple modalities, including 2D image data on a spatial metric space $\mathbb{R}^2$, 3D video data on a spatial-temporal metric space $\mathbb{R}^3$, and 3D viewpoint data on a camera pose and intrinsic parameter metric space $\mathbb{R}^6$, while the score network implementation remains identical across different modalities, except for the embedding size. The concrete network implementation details including architecture and hyper-parameters can be found in the appendix.

**Images.** For image generation, we use the standard benchmark dataset, *i.e.*, CIFAR10 64×64 Krizhevsky et al. (2009) as a sanity test, in order to compare with other domain-agnostic and domain-specific methods. For the low-resolution CIFAR10 dataset, we compare our method with the previous domain-agnostic methods including DPF Zhuang et al. (2023) and GEM Du et al. (2021). We report Fréchet Inception Distance (FID) Heusel et al. (2017) and Inception Score (IS) Salimans et al. (2016) or quantitative comparisons.

The experimental results can be found in Tab. 1. Specifically, T1 outperforms all domain-agnostic models in the FID and IS metrics. The qualitative comparisons in Fig. 4 further demonstrate our method's superiority in images. Note that our method does not use text descriptions for ensuring a fair comparison. It simply learns to predict all coordinate-signal pairs of a single image during training without using additional text descriptions or embeddings.

**Videos.** To show our model's capacity for more complex data, *i.e.*, high-resolution, dynamic video, we conduct experiments on the recent text-to-video benchmark: CelebV-Text 256×256×128 (Yu et al., 2023b) (128 frames). As additional spatial and temporal coherence is enforced compared to images, video generation is relatively underexplored by domain-agnostic methods. We compare our method with the representative domain-specific methods including TFGAN (Balaji et al., 2019), MMVID (Han et al., 2022a), CogVideo (Hong et al., 2023) and VDM (Ho et al., 2022b). We report Fréchet Video Distance (FVD) (Unterthiner et al., 2018), FID, and CLIPSIM (Wu et al., 2021), *i.e.*, the cosine similarity between the CLIP embeddings (Radford et al., 2021) of the generated

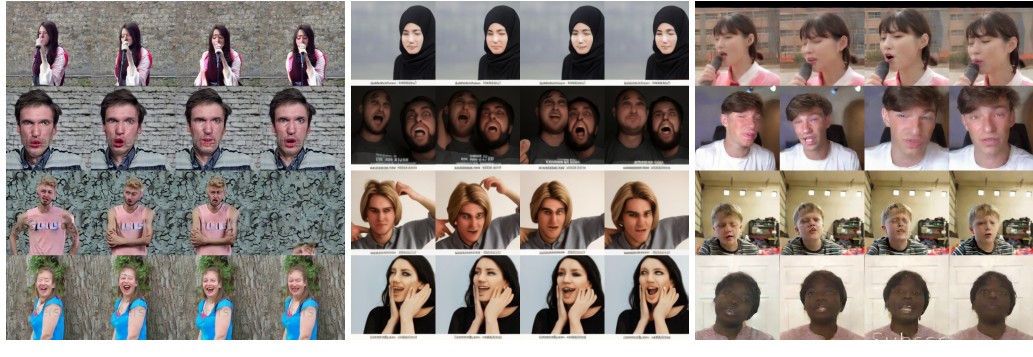

(a) VDM (Ho et al., 2022b)      (b) CogVideo (Hong et al., 2023)      (c) **T1** (Ours)

Figure 5: Qualitative comparisons of domain-specific text-to-video models and ours. Compared with VDM Ho et al. (2022b), our result is more continuous. Compared with CogVideo Hong et al. (2023), our result have more realistic textures. Please see `https://t1-diffusion-model.github.io/visual-comparisons-celebvt.html` for more results.

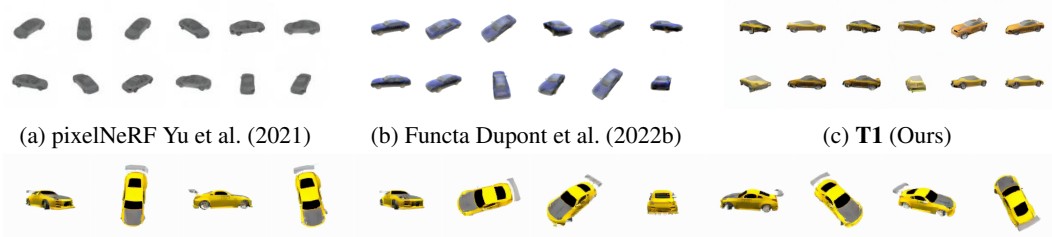

(a) pixelNeRF Yu et al. (2021)      (b) Functa Dupont et al. (2022b)      (c) **T1** (Ours)

(d) **T1** (Our high-resolution result)

Figure 6: Qualitative comparisons of domain-specific novel view synthesis models and ours. Our results show competitive quality without explicitly using 3D modeling and allows generating high-resolution results (*e.g.*, 256×256×251 ) by only using low-resolution training data. Please see `https://t1-diffusion-model.github.io/visual-comparisons-shapenet_3dviewpoints.html` for more results.

images and the corresponding texts. Note, the recent text-to-video models (like NUAW (Wu et al., 2022), Magicvideo (Zhou et al., 2022), Make-a-video (Singer et al., 2022), VLDM (Blattmann et al., 2023), etc.) are not included in our comparisons. This is solely because all of them neither provide implementation details, nor runnable code and pretrained checkpoints. Furthermore, their approaches are similar to VDM (Ho et al., 2022b), which is specifically tailored for video data.

Our method achieves the best performance in both the video quality (FVD) and signal frame quality (FID) in Tab. 1, compared with the recent domain-specific text-to-video models. Moreover, our model learns more semantics as suggested by the CLIPSIM scores. The results show that our model, as a domain-*agnostic* method, can achieve a performance on par with domain-*specific* methods in the generation of high-resolution, dynamic data. The qualitative comparisons in Fig. 5 further support our model in text-to-video generation compared with the recent state-of-the-art methods.

**3D Viewpoints.** We also evaluate our method on 3D viewpoint generation with the ShapeNet dataset (Chang et al., 2015). Specifically, we use the "car" class of ShapeNet which involves 3514 different cars. Each car object has 50 random viewpoints, where each viewpoint is in $128 \times 128$ resolution. Unlike previous domain-agnostic methods (Du et al., 2021; Zhuang et al., 2023) that model 3D geometry over voxel grids at $64^3$ resolution, we model over rendered camera views based on their corresponding camera poses and intrinsic parameters, similar to recent domain-specific methods (Sitzmann et al., 2019; Yu et al., 2021). This approach allows us to extract more view-wise coordinate-signal pairs while voxel grids only have 6 views. We report our results in comparison with the state-of-the-art view-synthesis algorithms including pixelNeRF (Yu et al., 2021),

| Text | View-wise Noise | Local Sampling | Resolution | Training Views | FVD ($\downarrow$) | FID ($\downarrow$) | CLIPSIM ($\uparrow$) | MACs | Mems |
|------|------|------|------|------|------|------|------|------|------|
| ✗ | N/A | ✓ | 16.0 | 8 | 608.27 | 34.10 | - | 113.31G | 15.34Gb |
| ✓ | ✗ | ✓ | 16.0 | 8 | 401.64 | 75.81 | 0.198 | 117.06G | 15.34Gb |
| ✓ | N/A | ✗ | 1.0* | 8 | 900.03 | 119.83 | 0.113 | 7.350T | 60.31Gb |
| ✓ | ✓ | ✓ | 1.0* | 8 | 115.20 | 40.34 | 0.187 | 7.314T | 22.99Gb |
| ✓ | ✓ | ✓ | 16.0 | 1 | 320.02 | 21.27 | 0.194 | 117.06G | 15.34Gb |
| ✓ | ✓ | ✓ | 16.0 | 4 | 89.83 | 23.69 | 0.194 | 117.06G | 15.34Gb |
| ✓ | ✓ | ✓ | 16.0 | 8 | 42.03 | 24.33 | 0.220 | 117.06G | 15.34Gb |

Table 2: Ablation analysis on our proposed method under different settings. '*' denotes that the model is trained on low-resolution $32\times32$ videos due the setting is not efficient enough and reach the memory constraints. All computation cost MACs and GPU memory usage Mems are estimated in generating a single view regardless of the resolution for a fair comparison.

viewFormer (Kulhánek et al., 2022), and EG3D-PTI (Chan et al., 2022). Note that our model indeed performs one-shot novel view synthesis by conditioning on the text embedding of a random view.

Our model's performance is even comparable with domain-*specific* novel view synthesize methods, as shown by the result in Tab. 1. Since our model does not explicitly utilize 3D geometry regularization as NeRF does, the compelling results demonstrate the potential of our method across various complex modalities like 3D geometry. The visualizations in Fig. 6 also show similar quality as previous works.

## 4.1 ABLATIONS AND DISCUSSIONS

In this section, we demonstrate the effectiveness of each of our proposed components and analyze their contributions to the quality of the final result, as well as the computation cost. The quantitative results under various settings are shown in Table 2. Since the text conditioning effect depends on our sampling algorithm, we will first discuss the effects of text conditions and then local sampling.

**Effect of text condition.** To verify the effectiveness of the text condition for capturing the global geometry of the data, we use two additional settings. **(1)** The performance of our model when the text condition is removed is shown in the first row of Tab. 2. The worse FVD means that the text condition play a crucial role in preserving the global geometry, specifically the spatial-temporal coherence in videos. **(2)** When the text condition is added, but not the cross-view consistent noise, the results can be found in the second row of Tab. 2. The FVD is slightly improved compared to the previous setting, but the FID is weakened due to underfitting against cross-view inconsistent noises. In contrast to our default setting, these results demonstrate the effectiveness of the view-consistent noise. Furthermore, we note that more detailed text descriptions can significantly improve the generated video quality.

**Effect of local sampling.** We investigate the effects of local sampling under different settings for preserving the local structure of data. **(1)** We first compare our local sampling with the baseline uniform sampling strategy (Zhuang et al., 2023), as shown in the 3rd row and 4th row of Tab. 2. Specifically, due to the memory constraints, we can only conduct experiments on frames in a lower resolution of $32\times32$ during sampling pairs, which are marked with "*". The FID evaluated on single frames shows the local structure quality, and hence the effectiveness of local sampling. Furthermore, our local sampling significantly reduces memory usages, from 60.31Gb into 22.99Gb, at a 0.036T less cost of MACs. **(2)** To verify the effectiveness of the extended signal resolution, we can compare the 4th row (resolution $1\times1$) and the last row (default setting; resolution $16\times16$). In contrast, our default setting outperforms the low-resolution setting without significant computation and memory consumption.

**Effect of number of views.** We investigate the model performance change with varying number of views ($n$) for representing fields, as shown in the 5th and 6th rows of Tab. 2. Compared to the default setting of $n = 8$, reducing $n$ to 1 leads to non-continuous frames and abrupt identity changes, as indicated by the low FVD. When $n$ is increased to 4, the continuity between frames is improved, but still worse than the default setting with $n = 8$ for the dynamics between frames. As the $n = 8$ setting reaches the memory limit, we set it as the default. Thus, a larger number of views leads to a higher performance, along with a higher computation cost.

**Effect of network architecture.** Different from DPF (Zhuang et al., 2023) that depends on context-query pairs, our model can also utilize the non-transformer architecture like the U-Net architecture used in LDM Rombach et al. (2022). To demonstrate our flexibility, we include a supplemental 3D-viewpoints generation experiment by replacing the transformer architecture with the U-Net network and self-attention used in LDM, where the coordinates are embedded as an additional channel of the latent. Compared with the used transformer architecture that achieves 24.36 FID, the new network can archives comparable 26.92 FID performance, which still matches the domain-specific methods.

**Limitations.** (1) Our method can generate high-resolution data, but the scaling property is merely resolved for the spatial dimensions exclusively. For instance, for an extremely long video with complex dynamics (*e.g.*, 1 hour; such long videos remain uncharted in the literature), learning short-term variations is still difficult since our local sampling method is still uniform in the temporal perspective. This paper focuses on generating spatially high-resolution data. (2) Our method only applies to visual modalities interpretable by views. For modalities such as temperature manifold (Hersbach et al., 2019) where there is no "views" of such field, our method does not apply. As long as the data in the new domain (e.g., 3D dynamic scene and MRI) can be interpreted by views, our method can reuse the same latent autoencoder (Rombach et al., 2022) without switching to domain-specific autoencoders.

## 5 RELATED WORK

In recent years, generative models have shown impressive performance in visual content generation. The major families are generative adversarial networks (Goodfellow et al., 2020; Mao et al., 2017; Karras et al., 2019; Brock et al., 2018), variational autoencoders (Kingma & Welling, 2013; Vahdat & Kautz, 2020), auto-aggressive networks (Chen et al., 2020; Esser et al., 2021), and diffusion models Ho et al. (2020); Song et al. (2020). Recent diffusion models have obtained significant advancement with stronger network architectures (Dhariwal & Nichol, 2021), additional text conditions (Ramesh et al., 2022), and pretrained latent space (He et al., 2022). Our method built upon these successes and targets at scaling domain-agnostic models for matching these advancement.

Our method models field distributions using explicit coordinate-signal pairs, which is different from the body of work that implicitly models field distributions, including Functa (Dupont et al., 2022b) and GEM (Du et al., 2021). These methods employ a two-stage modeling paradigm, which first parameterizes fields and then learns the distributions over the parameterized latent space. Compared with the single-stage parameterization used in our method, the two-stage paradigm demands more complex network architecture, as it employs a separate network to formulate a hypernetwork (Ha et al., 2016). Moreover, the learning efficiency of the two-stage methods hinders scaling the models, as their first stage incurs substantial computational costs to compress fields into latent codes. In contrast, our method enjoy the benefits of the single-stage modeling and improves its accuracy in preserving local structures and global geometry.

Our method also differs from the recently proposed domain-specific works for high-resolution, dynamic data, which models specific modalities in a dedicated latent space, including Spatial Functa (Bauer et al., 2023) and PVDM (Yu et al., 2023c). These methods typically compress the high-dimensional data into a low-dimensional latent space. However, the compression is usually specific to a center modality and lacks the flexibility in dealing with different modalities. For instances, PVDM compresses videos into three latent codes that represent spatial and temporal dimensions separately. However, such a compressor cannot be adopted into the other similar modalities like 3D scenes. In contrast, our method owns the unification flexibility by learning on the coordinate-signal pairs and the achieved advancement can be easily transferred into different modalities.

## 6 CONCLUSION

In this paper, we introduce a new generative model to scale the DPF model for high-resolution data generation, while inheriting its modality-agnostic flexibility. Our method involves (1) a new view-wise sampling algorithm based on high-resolution signals; (2) a conditioning mechanism that leverages view-level noise and text descriptions as inductive bias. Experimental results demonstrate its effectiveness in various modalities including image, video, and 3D viewpoint.

## 7 ETHICAL STATEMENT

In this paper, we present a new generative model unifying varies visual content modalities including images, videos, and 3D scenes. While we are excited about the potential applications of our model, we are also acutely aware of the possible risks and challenges associated with its deployment. Our model's ability to generate realistic videos and 3D scenes could potentially be misused for creating disingenuous data, *a.k,a*, "DeepFakes". We encourage the research community and practitioners to follow privacy-preserving practices when utilizing our model. We also encourage readers to refer to the Rostamzadeh et al. (Rostamzadeh et al., 2021) for an in-depth review of ethics in generating visual contents.

## 8 REPRODUCIBILITY STATEMENT

We provide the hyperparameters of each presented experiments in Appendix B. Readers can reproduce our results according to our listed implementation details, including the dimension, number of channels, and data resolution in Appendix B. Furthermore, we also provide the training and testing data details in Appendix C for reference.

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

# A    ADDITIONAL RESULTS

- More text-to-video generation result comparisons: `https://t1-diffusion-model.github.io/visual-comparisons-celebvt.html`
- More non-face text-to-video generation results: `https://t1-diffusion-model.github.io/visual-comparisons-webvid.html`
- More 3D viewpoints generation results: `https://t1-diffusion-model.github.io/visual-comparisons-shapenet_3dviewpoints.html`
- More 3D voxel[1] generation result comparisons: `https://t1-diffusion-model.github.io/visual-comparisons-shapenet_voxels.html`

# B    ADDITIONAL SETTINGS

**Model Details.**

- In the interest of maintaining simplicity, we adhere to the methodology outlined by Dhariwal et al. (Dhariwal & Nichol, 2021) and utilize a 256-dimensional frequency embedding to encapsulate input denoising timesteps. This embedding is then refined through a two-layer Multilayer Perceptron (MLP) with Swish (SiLU) activation functions.
- Our model aligns with the size configuration of DiT-XL (Peebles & Xie, 2022b), which includes retaining the number of transformer blocks (*i.e.* 28), the hidden dimension size of each transformer block (*i.e.*, 1152), and the number of attention heads (*i.e.*, 16).
- Our model derives text embeddings employing T5-XXL (Raffel et al., 2020), culminating in a fixed length token sequence (*i.e.*, 256) which matches the length of the noisy tokens. To further process each text embedding token, our model compresses them via a single layer MLP, which has a hidden dimension size identical to that of the transformer block.

**Diffusion Process Details.**    Our model uses classifier-free guidance in the backward process with a fixed scale of 8.5. To keep consistency with DiT-XL (Peebles & Xie, 2022a), we only applied guidance to the first three channels of each denoised token.

**3D Geometry Rendering Settings.**    Following the settings of pixelNeRF (Yu et al., 2021), we render each car voxel into 128 random views for training models and testing. However, the original setting puts camera far away from the objects and hence results in two many blank areas in the rendered views. We empirically find that these blank areas hurts the diffusion model performance since the noise becomes obvious in blank area and can be easily inferred by diffusion models, which degrades the distribution modeling capability of diffusion models.

To overcome this, we first randomly fill the blank area with Gaussian noise $\mathcal{N}(0, 0.1)$ without overlapping the 3D geometry. We then move the camera in the z-axis from 4.0 into 3.0, which is closer to the object than the previous one. During testing, we use the same settings as pixelNeRF and remove the noise according to the mask. For straightforward understand their difference, we visualized their rendered results in Fig. 7.

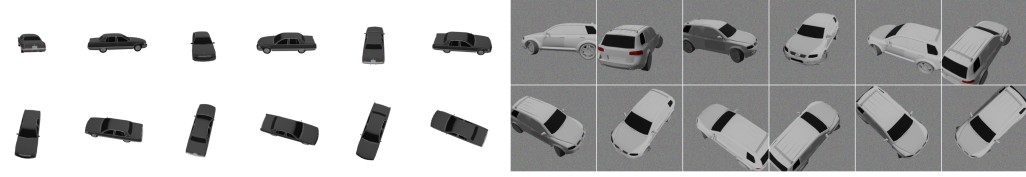

(a) pixelNeRF Yu et al. (2021) rendering                    (b) ours rendering

Figure 7: Visual comparisons of different 3D geometry rendering settings.

---

[1]Note that the comparisons are conducted by rendering their generated voxel objects into 3D viewpoints and then compare those results with ours.

| Hyper-parameter | CIFAR10 (Krizhevsky et al., 2009) | CelebV-Text (Yu et al., 2023b) | ShapeNet (Chang et al., 2015) |
|---|---|---|---|
| train res. | 64×64 | 256×256×128 | 128×128×128
256×256×128 (upsampled) |
| eval res. | 64×64 | 256×256×128 | 128×128×251
256×256×251 |
| # dim coordinates | 2 | 3 | 6 |
| # dim signal | 3 | 3 | 3 |
| # freq pos. embed | 10 | 10 | 10 |
| # freq pos. embed $t$ | 64 | 64 | 64 |
| #blocks | 28 | 28 | 28 |
| #block latents | 1152 | 1152 | 1152 |
| #self attention heads | 16 | 16 | 16 |
| batch size | 128 | 128 | 128 |
| lr | $1e-4$ | $1e-4$ | $1e-4$ |
| epochs | 400 | 400 | 1200 |

Table 3: Hyperparameters and settings on different datasets.

## C  ADDITIONAL DATASET DETAILS

In the subsequent sections, we present the datasets utilized for conducting our experiments. We empirically change the size settings of our model as shown in Tab 3.

- **CelebV-Text** (Yu et al., 2023b) Due to the unavailability of some videos in the released dataset, we utilize the first 60,000 downloadable videos for training our model. For videos that contain more than 128 frames, we uniformly select 128 frames. Conversely, for videos with fewer than 128 frames, we move to the next video, following the order of their names, until we identify a video that meets the required length of 128 frames.

- **ShapeNet (Chang et al., 2015).** The conventional methods in DPF (Zhuang et al., 2023) and GEM (Du et al., 2021) generally involve training models on the ShapeNet dataset, wherein each object is depicted as a voxel grid at a resolution of $64^3$. However, our model distinguishes itself by relying on view-level pairs, thereby adopting strategies utilized by innovative view synthesis methods like pixelNeRF (Yu et al., 2021) and GeNVS (Chan et al., 2023). To specify, we conduct training on the car classes of ShapeNet, which encompasses 2,458 cars, each demonstrated with 128 renderings randomly scattered across the surface of a sphere.

  Moreover, it's worth noting that our model refrains from directly leveraging the text descriptions of the car images. Instead, it conditions on the CLIP embedding (Radford et al., 2021) of car images for linguistic guidance. This approach circumvents the potential accumulation of errors that might occur during the text-to-image transformation process.

## D  ADDITIONAL EXPERIMENTAL DETAILS

**Video Generation Metrics Settings.**  In video generation, we use FVD (Unterthiner et al., 2018)[2] to evaluate the video spatial-temporal coherency, FID (Heusel et al., 2017)[3] to evaluate the frame quality, and CLIPSIM (Radford et al., 2021)[4] to evaluate relevance between the generated video and input text. As all metrics are sensitive to data scale during testing, we randomly select 2,048 videos from the test data and generate results as the "real" and "fake" part in our metric experiments. For FID, we uniformly sample 4 frames from each video and use a total of 8,192 images. For CLIPSIM, we calculate the average score across all frames. We use the "openai/clip-vit-large-patch14" model for extracting features in CLIPSIM calculation.

---

[2]FVD is implemented in `https://github.com/sihyun-yu/DIGAN`

[3]FID is implemented in `https://github.com/toshas/torch-fidelity`

[4]CLIPSIM is implemented in `https://github.com/Lightning-AI/torchmetrics`

# E VISUALIZATION

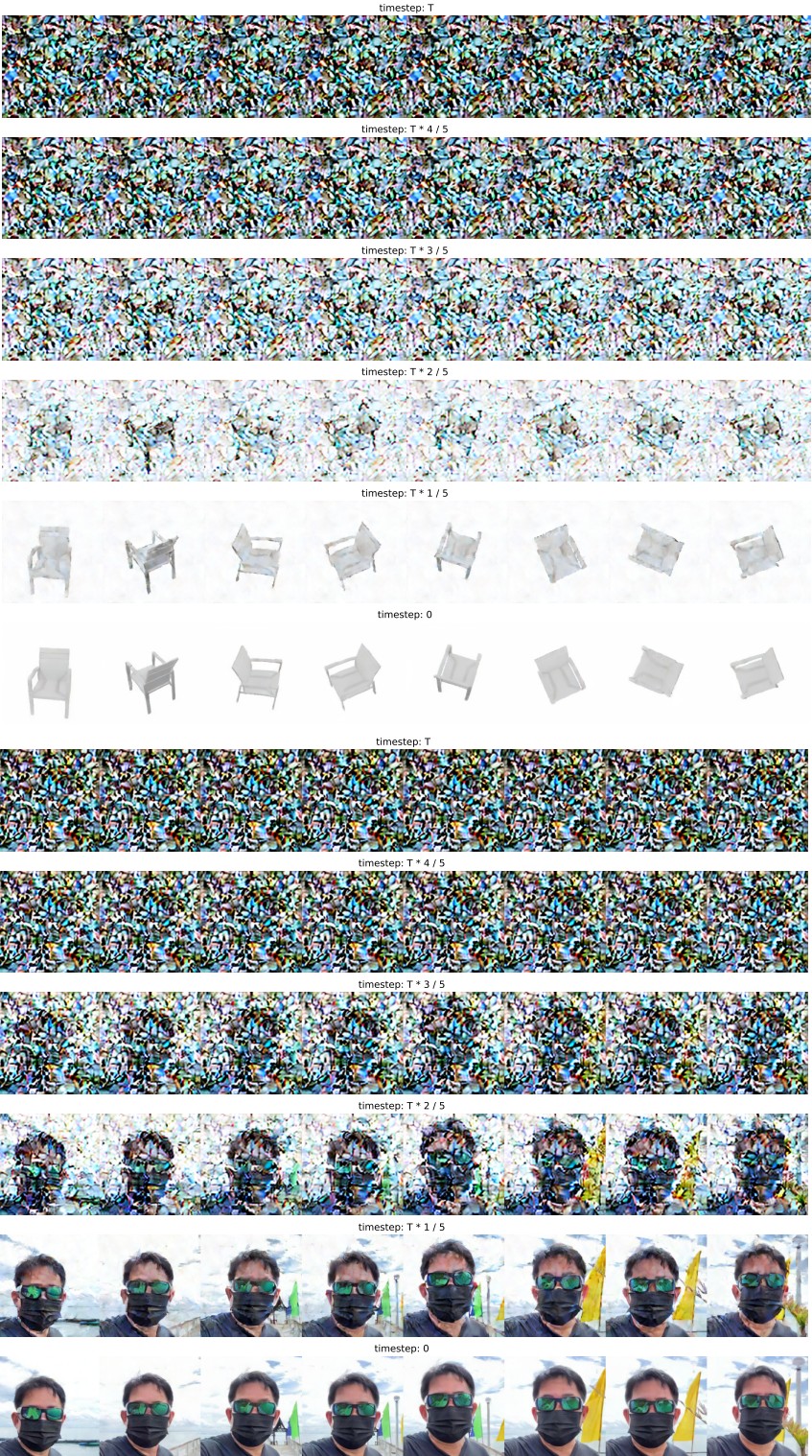

Figure 8: Intermediate results visualization of our multi-view generation. Here we show the predicted clean image at different timesteps for highlighting the cross-view consistency of our model.

