# OpenReview forum: "T1: Scaling Diffusion Probabilistic Fields to High-Resolution on Unified Visual Modalities"
_ICLR.cc/2024/Conference — ICLR 2024 Conference Withdrawn Submission_

### Official Review · Reviewer_DYLH · 2023-10-30

**Soundness:** 2 fair
**Presentation:** 3 good
**Contribution:** 2 fair
**Rating:** 5
**Confidence:** 4

**Summary:**

The authors proposed upscaling the diffusion probabilistic fields to higher resolutions. To achieve this, they introduced a visual content tokenization method, which involves representing each patch as tokens rather than treating signals as independent points. Following content representation, a decoder-only network is employed for field generation. Additionally, shared noise across multiple views and text conditioning are integrated (for 3D shapes and videos) to maintain global consistency.

**Strengths:**

The paper maintains overall clarity. The results outperform those of the baseline methods. The topic of upscaling field generation is interesting and impactful.

**Weaknesses:**

To reduce resource costs, the authors suggest tokenizing the patch input as a compression representation. While this approach performs admirably on image datasets, it appears challenging to maintain global consistency on video datasets. Moreover, given the recent release of advanced 3D datasets like Objaverse, it will be impactful to conduct experiments on these complex datasets. Details are listed below.


Results:
1) The claimed global consistency of the proposed method appears overstated. Flickering issues are readily discernible in all the video results, whether involving faces or general scenes, as seen in the supplementary videos.

2) Building on the previous point, the authors assert that using shared noise and text conditioning aids in improving global consistency. However, this improvement is not clearly observed in the video results. The ablation studies on shared noise and text conditioning are inadequate. Additionally, further explanation or analysis of the flickering phenomenon is encouraged.

3) The current method still employs uniform sampling to me, but it's over patches rather than single pixels or voxels. In this case, the authors are opting for input compression rather than proposing more advanced sampling methods, although random sampling is employed for tokens in the case of video datasets.. The sampling process is still over the entire metric space, which may be considered inefficient. For instance, when representing 3D shapes through occupancy, a significant portion of the space remains unoccupied.


Miscellaneous:
1) The writing of the View-wise Sampling is challenging to follow. Furthermore, implementation details of the sampling process are absent, especially on different datasets (e.g., on 3D shapes).
2) The writing could benefit from better organization. For instance, there are inconsistencies in phrase usage for the same concept (if I understand correctly), such as "view-wise sampling" in Section "Method" and "local sampling" in the Tables.

**Questions:**

1) How do the baseline methods perform on video datasets like CelebV-Text (even at a low resolution), particularly those that do not incorporate the proposed tokenization method?

2) The current method still employs uniform sampling, but over patches rather than pixels or voxels. I'm curious if there's an advanced sampling strategy for selectively sampling non-empty regions. This could be highly impactful, particularly when sampling meaningful regions, for instance, on 3D shapes and their representations.

3) In Table 2, what does "N/A" refer to, as it is distinct from the checkmark and cross mark indicators?

---

### Official Review · Reviewer_5j9d · 2023-10-31

**Soundness:** 2 fair
**Presentation:** 3 good
**Contribution:** 3 good
**Rating:** 3
**Confidence:** 5

**Summary:**

This paper presents a new unified data generation framework, which is based on Diffusion Probabilistic Field (DPF). Considering the difficulty for DPF to capture local structures and can hardly scale to higher data resolution, the authors introduce a view-wise sampling algorithm to learn local structure. In addition, text description guidance is incorporated to complement the global geometry. Based on the above insights, the proposed model can be scaled to generate high-resolution data while unifying multiple modalities. Experimental comparisons on data generation in various modalities demonstrate the effectiveness of our model.

**Strengths:**

1. The motivation and implementation of the proposed method is well illustrated and involves sufficient generation experiments on various tasks and datasets.
2. A novel view-wise sampling algorithm is proposed to enable local structure capture.
3. The presented model operates in the latent space of the autoencoder instead of pixel space, allowing to deal with data views in higher resolution when take the same number of pairs as input.
4. Experimental results, including comparisons with the baseline methods and ablation studies, further demonstrate the effectiveness of this method.

**Weaknesses:**

1. As stated in Section “Signal Resolution” of page 4, “the method can handle views with a maximum resolution of 1024 × 1024 while maintaining very close memory consumption”, however, the experiments of the paper only involve data generation of 128 × 128 and 256 × 256 resolutions, which is not enough to verify the statement in this section.
2. Since the authors emphasize the trade-off between high resolution data generation and hardware capacity, it is necessary to add metric comparisons with existing methods regarding model efficiency in the quantitative experiments section, e.g., Table 1.
3. Qualitative comparisons in Figure 5 are not sufficient to illustrate the validity of the proposed method for the following reasons: 1) Without providing the input text description, it is difficult to determine the generation consistency of the video results and the text description; 2) Upon careful observation and comparison of the provided video results, it is found that although the method in this paper is able to synthesize relatively natural textures and head dynamics, there is significant facial jitters around facial areas, as shown in the second line of comparison with the VDM method in Figure 5; 3) The fairness of the comparisons need to be further verified, the method in this paper is trained on the face dataset CelebV-Text whereas the VDM is trained on a dataset of human action, does the bias in the training set and the mismatch between datasets and the input text description lead to a biased generation effect?

**Questions:**

1. The description of Figure 1(c) is missing from the introduction section.
2. The equation of the adLNorm in page 6 seems to have the wrong symbols.
3. The optimal results in Table 2 for quantitative comparisons are suggested to be bolded.

---

### Official Review · Reviewer_3ac9 · 2023-11-04

**Soundness:** 3 good
**Presentation:** 3 good
**Contribution:** 2 fair
**Rating:** 3
**Confidence:** 4

**Summary:**

This paper presents an enhancement of Diffusion Probabilistic Field (DPF) models, a unified generative model designed for multiple modalities. The proposed framework, named T1, extends the capabilities of DPF by enabling the generation of high-resolution 256x256 videos, surpassing DPF's 64x64 image generation limit. This achievement is made possible through the incorporation of two key components: a view-wise sampling module and additional text conditioning.

T1's performance was evaluated across three benchmark datasets: CIFAR10 for 64×64 image generation, CelebV-Text for 256×256×128 video generation, and ShapeNet-Cars for 128×128×251 view-synthesis. The results demonstrate that T1 outperforms several baseline models, showcasing its efficacy in various generative tasks.

**Strengths:**

This study delves into a unified diffusion model that spans across multiple modalities, offering several notable strengths:

1. **Straightforward Design**: The incorporation of view-wise sampling is an intuitive design choice. It synergizes two well-established components: an encoder for generating latents from input data and DiT as the diffusion backbone. This thoughtfully designed approach empowers T1 to achieve scalability.

2. **Effective Text Conditioning**: The inclusion of text conditioning is both reasonable and effective, enhancing the model's performance.

3. **Promising Experimental Results**: The experimental findings are promising, with T1 consistently outperforming baseline models by great margins.

4. **Clarity in Communication**: The writing in this work is mostly clear and easy to follow, which enhances the overall presentation and accessibility of the research.

**Weaknesses:**

While this paper explores an intriguing research direction and demonstrates promising results, there are several notable weaknesses, in my opinion:

1. **Limited Technical Contribution**: The technical contribution appears somewhat limited. The use of the encoder, a key component, is well-established in LDM/SD, and text conditioning has been validated in numerous prior works.

2. **Quality of Generated Content**: The quality of the generated images and videos falls short of the state-of-the-art in single-modality generative models. Notably, the generated videos exhibit numerous artifacts, undermining the persuasiveness of the results.

3. **Changes in 3D Experiment**: The authors' decision to change the original 3D experiment in DPF from volumetric reconstruction to view-synthesis raises concerns. Firstly, reconstruction seems more like a different modality compared to view-synthesis in the context of image/video generation. Additionally, the choice of newer and stronger baselines, as opposed to PixelNeRF, would bolster the credibility of the results. PixelNeRF is designed for few-shot NeRF and bit outdated.

4. **Incomplete Related Work Section**: The related work section lacks references to several relevant works, including:
   - "Deep unsupervised learning using nonequilibrium thermodynamics, ICML'15," which is a pioneering diffusion model paper.
   - The omission of class-specific models for image/video/3D generation.
   - Neglecting to mention multi-modality research in other fields of computer vision and machine learning.

5. **Confusing Approach Section**: The writing in the approach section is confusing and could benefit from clarification. For example, Section 3.1 is titled "view-wise sampling algorithm," but the subsections "Signal Resolution" and "Score Network" appear to be more related to architectural design. Similarly, the subsections "Cross-view Noise Consistency" and "Cross-view Condition Consistency" may not be appropriately categorized under "text conditioning."

**Miscellaneous Issues**:
- The appendix provides only hyperparameters and some implementation details. The absence of source code is a notable shortcoming.

**Questions:**

While reviewing the paper, some important details are not clear:

1. How is the text data for image and 3D data generated? The original cifar10 and shapenet don't contain text annotation. If these text descriptions are generated automatically, how does this design choice influence the final results?

2. Page 4 it's stated that `our method can handle views with a maximum resolution of 1024 × 1024 while maintaining very close memory consumption during learning without compromising the quality of the generated signal`. Where can I find the 1024 × 1024 results?

3. What's the motivation for using adLNorm for text? What's the advantage of this design choice over alternatives like cross-attention.

---

### Official Review · Reviewer_DQqx · 2023-11-09

**Soundness:** 2 fair
**Presentation:** 2 fair
**Contribution:** 2 fair
**Rating:** 3
**Confidence:** 3

**Summary:**

The paper presented a method to scale up diffusion probabilistic fields (DPF) for generating high resolution visual data (e.g., image or videos). The proposed method incorporated several techniques, including a reparameterization to downsample the input metric space, a variant of the score network, a view-wise sampling strategy, and text conditioned generation. The proposed method was demonstrated on three image / video generation tasks. The results are encouraging.

**Strengths:**

* The paper seeks to scale up DPF, with the potential to support a large range of generative tasks.

* The experiments are extensive and the results are promising.

**Weaknesses:**

The vanilla DPF presents a method to model distribution over fields, where a field is defined as a mapping between an input metric space and an output metric space. This formulation is rather general and can be used for many generative tasks. The proposed method, however, seems to have made certain assumptions to simplify DPF in order to scale it up. Unfortunately, these assumptions were not clearly described in the paper.

* A key assumption lies in the definition of views, central to the proposed method. This assumes that input metric space can be organized into views. While this is possible for many image/video generation tasks (say video generation where a frame is a view), it precludes some other visual tasks. For example, the generation of 3D shapes as described in DPF, where the field maps from R^3 to R with no clear definition of “views.” The authors indeed recognized this as one of the limitations, yet it was not described till the end of Section 4.

* Another assumption is that the input / output space can be reparameterized into a lower resolution signal. In the paper, this is achieved using a VQ-VAE (as stable diffusion). Again, this is fine for image / video generation. Yet for an arbitrary field, such reparameterization might not be possible. Consider the setting of modeling 3D viewpoints (NeRF type of models) in Fig. 1 (c). The input space is R^5 (x, y, z in 3D and viewing angles) and output space is R^4 (color and density). Contrary to what the paper stated, I don’t think there is an easy way to apply an autoencoder in this case to reduce the input resolution. One can train an autoencoder in the output space, yet that does not help to subsample the input, as the "views" are no organized in the input space. Can the authors further clarify?

* The method also seems to assume that the input metric space must be discrete, by stating that “the context pairs are no longer necessary.” Even if a dense set of tokens (defined on grids, after autoencoder) capture sufficient information about the field, if the input space is continuous, these context pairs will still be needed to interpolate signals between the grids. Again, I will refer back to the 3D viewpoints (NeRF) example, which requires a continuous input metric space. I don’t think the proposed model can handle the continuous input space, unless I missed something.

If I understand correctly, the view-wise sampling essentially “factorizes” a field into a set of fields defined on each view (or a subset of views). Provided that each view is an image and that context pairs are not used, how is the method (for individual views) different from a diffusion model for image generation? Yes, the pixel coordinates are used as part of the input, but they seem to no longer influence the generation process.

Part of the paper, especially the technical components, were written in a somewhat opaque manner. Implementation details were mixed with technical concepts. Key assumptions were not explained till much later in the paper. These issues have made the paper hard to comprehend.

**Questions:**

See my questions in the weaknesses section.